**Data Availability Statement:** All relevant data are within the manuscript and its Supporting Information files.

**Funding:** The authors received no specific funding for this work.

# Platelet-rich plasma for rotator cuff tendinopathy: A systematic review and meta-analysis

**Mohamad Shariff A. Hamid[1]**, **Shariff Ghazali Sazlina**[2,3]*

1 Faculty of Medicine, Unit of Sports Medicine, University of Malaya, Kuala Lumpur, Malaysia, 2 Faculty of Medicine and Health Sciences, Department of Family Medicine, Universiti Putra Malaysia, Serdang, Selangor, Malaysia, 3 Malaysian Research Institute on Ageing (MyAgeing™), Universiti Putra Malaysia, Serdang, Selangor, Malaysia

☯ These authors contributed equally to this work.

* sazlina@medic.upm.edu.my

## Abstract

Shoulder rotator cuff tendinopathy is a common debilitating condition that affects a person daily function and the quality of life. Despite its frequent occurrence, the best treatment is still inconclusive. This review assessed the clinical effect of platelet-rich plasma injection for rotator cuff tendinopathy. A systematic literature search was conducted using CINAHL, Medline, SCOPUS, SPORTSDiscus and Web of Science databases to retrieve articles published in peer-reviewed journals until December 2020. Randomised controlled trials (RCTs), which compared clinical effects of PRP injection to the usual care among adults diagnosed with rotator cuff conditions were reviewed. The main outcomes of interest were changes in shoulder pain symptoms and shoulder functions. All variables were analysed using random effects meta-analyses. Eight RCTs were reviewed in this study. The risk of bias for randomization was low for 6 RCTs, one study had unclear risk and the other was a high risk. Studies vary on the PRP techniques including preparation and injections. Moreover, the control intervention also differs. Four studies compared PRP with normal saline injection while in the remaining 4 RCTs the control intervention were rehabilitation program and dry needling. Meta-analysis of selected studies showed that PRP injection was safe and effective intervention for long-term pain control and shoulder function in patients with RC disorders.

## Introduction

In the healthcare setting, shoulder pain and dysfunction are commonly observed. Epidemiological studies have shown a wide prevalence of these disorders varying from 7% to 26% [1]. The shoulder disorders have been identified as the third most common musculoskeletal complaint, according to a large population study in the United Kingdom [2]. Shoulder disorders have a major effect on the capacity of a person to perform daily activities and on the quality of life [3]. Among shoulder pathologies, rotator cuff tendinopathy is a common cause of shoulder pain and functional deficit. Studies in primary care settings have shown that the incidence of

**Competing interests:** The authors have declared that no competing interests exist.

rotator cuff pathologies were 15%-50% [4,5]. A recent systematic review found that it took 10 months for patients diagnosed with shoulder tendinopathy to fully recover. Also, during the recovery phase, patients were reported to be less productive, took many days of sick leave, and require compensation for the condition [6].

The rotator cuff (RC) is composed of 4 muscles; the supraspinatus, infraspinatus, teres minor and subscapularis whose distal tendon attached tubercles of the humerus. The primary function of the RC is to hold the humeral head in the glenoid cavity of the scapula during all movements of the glenohumeral joint [7]. A healthy rotator cuff is a pain-free, structurally-organized tendon capable of performing routine functional tasks [8]. Tendinopathy is described as an overuse disorder characterised by pain in and around the tendon with impaired tendon function [9,10]. Patient accounts of pain and impairment are the most critical features of tendinopathy. A self-reported questionnaire is a method for measuring and monitoring the degree of pain, impairment and disease progression [10]. In the diagnosis of RC tendinopathy, clinical examination showed moderate sensitivity of about 70 percent [11]. Radiological imaging, including shoulder MRI and ultrasound is sensitive to detect structural tendinopathic changes. However, because MRI is costly, shoulder ultrasound has become a standard method to test tendon integrity with 0.84 and 0.89 sensitivity and specificity, respectively [12].

Treatment options for RC tendinopathy vary from conservative, noninvasive, including a recovery program, to more invasive procedures such as shoulder injections and surgery. A systematic review showed limited evidence that surgery is more effective in treating rotator cuff tear than conservative treatment alone [13]. In the early inflammatory phase of tendon disease, anti-inflammatory agents, rehabilitation exercise program and electrotherapeutic modalities are often used for symptom control [14]. However, the management of recalcitrant tendinopathies is difficult despite numerous treatment options. Once it is weakened, the biological and biomechanical properties of tendon tissue are never completely restored. Healing times are prolonged in chronic tendinopathies because tendons are comparatively less vascular and local blood flow to the muscles is less than regulated. Thus, the use of platelet-rich plasma (PRP) and prolotherapy has recently emerged as a treatment choice for rotator cuff tendinopathy [15]. The supraphysiological platelet concentration that can release biologically active proteins that facilitate cell recruitment, development and morphogenesis is the premise behind the PRP used. Increased fibroblast migration and proliferation, increased tissue vascularisation and collagen deposition are shown to stimulate tendon healing [16].

To date, there are limited randomised clinical trials that have investigated the effects of PRP injection on RC injuries. The available RCTs differ in the study design and several studies compared the clinical effects of PRP injection with other forms of injections (e.g., corticosteroids or prolotherapy), while others use non-injections approach as comparison controls [17–19]. Although several studies have demonstrated the clinical value on the use of PRP for rotator cuff injuries, there is no consensus in the literature, as other studies have shown no definite benefit of PRP over corticosteroid injection or physical therapy. The goal of this systematic review was to scientifically assess trials using PRP as a non-surgical treatment for rotator cuff tendinopathy. The effects of the intervention on pain and shoulder function compared with control were evaluated in this review. Also, side effects of PRP treatment were investigated.

## Materials and methods

A systematic review and meta-analysis were conducted to explore the roles of PRP in managing the rotator cuff injuries. The review question was how effective are PRP in reducing shoulder pain and improving shoulder function?" This review was registered with the International

prospective register of systematic reviews (PROSPERO) CRD42018093081 (https://www.crd.
york.ac.uk/prospero).

## Study selection

This review was conducted according to the Preferred Reporting Items for Systematic reviews
and Meta-Analyses (PRISMA) guidelines [20,21]. Only randomised controlled trials (RCT)
were considered in this review. We included studies that compared PRP injection among
adults diagnosed with rotator cuff conditions to usual care. No restrictions were defined
regarding the type and content of the control group. The comparative interventions include
one or a combination of 1) normal saline, 2) no injection and 3) shoulder rehabilitation pro-
gram. In addition, RCTs with more than 2 arms were included if one of the arms fulfills criteria
of control intervention mentioned. The primary outcome measure in the selected studies was
change in pain symptoms. The secondary outcome measured was changes in shoulder func-
tion. Shoulder function assessed using various forms of questionnaires as well as assessment of
shoulder range of movement were considered in this review. The process of this search method
included describing the data sources, search strategy, data extraction and quality assessment.
The supporting PRISMA checklist is available as supporting information; See Checklist S1
File.

## Data sources and search strategy

Studies were searched electronically using the CINAHL, Medline, SCOPUS, SPORTSDiscus
and Web of Science databases. The search strategy was performed using the Medical Subject
Headings (MeSH) terms and keywords. For the rotator cuff, search was performed using the
MeSH term rotator cuff. For PRP, we used a combination of MeSH terms as follows: platelet-
rich plasma OR platelet-rich fibrin OR preparation rich in growth factors OR autologous con-
dition plasma OR autologous blood OR platelet concentrate OR platelet gel OR autologous
growth factors OR platelet realasate. A full electronic search strategy for Scopus database is
presented in S2 File. The reference lists of review articles and included studies were hand
searched for other potentially eligible studies using the same selection criteria as described
above. Articles published in the peer-reviewed journal until December 2020 were searched.
Due to limited sources, only papers published in English were considered. Original authors
were contacted twice (2 months apart) via email for data relevant to the meta-analysis. Equiva-
lent searches were performed on the other databases. Also, a search for archived articles from
the South East Asian region using similar selection criteria was performed through the local
library websites.

## Data extraction

The titles and abstracts of all articles retrieved from the search were reviewed based on the
selection criteria for the study to determine whether the full text manuscripts were required
for further evaluation. Each manuscript was systematically analysed according to the study's
objectives (on the effectiveness of interventions), research characteristics (study design, partic-
ipant's age and sample size), features of intervention (intervention strategies, intervention pro-
vider, length of intervention and participants' follow-up schedule), targeted outcome/s and
major findings.

### Methodological quality assessment of individual studies

Each selected study was evaluated for its methodological quality using the Cochrane Collaboration tool to assess the risk of bias (the Cochrane Handbook for Systematic Review of Interventions) [22]. The tool assessed studies; a) sequence generation, b) allocation concealment, c) blinding of participants, d) completeness of outcome data (including participant attrition), e) selective outcome reporting, and f) other areas of bias. For each domain, the procedures performed for each study were described and rated as 'high', 'low' or 'unclear' risk of bias based on gathered information. Two review authors made these judgements independently on the basis of predetermined criteria and discussed them in a meeting until a consensus was achieved.

### Data synthesis and analyses

We conducted a narrative synthesis based on the primary and secondary outcomes of this review. The primary outcome measures were pooled and calculated using the statistical software RevMan 5.3, according to the Cochrane Handbook for Systematic Reviews of Interventions [22]. Attempts to contact the authors to obtain the raw data for data analysis were made, of which 3 of 6 authors responded and provided raw data. The results of the BMI z-score, waist circumference, and body fat percentage were analysed using weighted or standardised mean differences as a measure of effect size, with 95% confidence intervals. Since the participant demographics and clinical settings differed among studies; we assumed the presence of heterogeneity a priori. Therefore, we used a random effects model to pool the results. We assessed heterogeneity using Cochran's Q statistic of heterogeneity with reported p-value and the degree of inconsistency across studies was quantified using I. In studies with 4 arm RCTs, each intervention group was independently analysed and compared with the control group. A funnel plot was performed to determine the presence of potential publication bias using the statistical software RevMan 5.3.

## Results

### Search results

A total of 548 articles were identified through the 5 databases and cross referencing.

Twenty-eight duplicates were identified and removed from the list. Upon screening titles and abstracts, 26 full-text articles were retrieved and assessed for eligibility. Eighteen articles did not fulfill the selection criteria and were excluded. The reasons for exclusion include a non-RCT study (n = 7), intervention was performed as adjunct to surgical procedures (n = 6), and using other forms of substance (n = 5). Eight articles were included in the narrative synthesis and meta-analyses. The flow diagram for selecting the studies is described in Fig 1.

### Characteristics of selected studies

The characteristics of the selected studies are summarised in Table 1. Two RCTs were conducted in Turkey [17,23]. The remaining 6 were performed in Canada [24], China [25], Iran [26], Switzerland [27], South Korea [28], and the United States of America [29]. All studies were conducted at the hospital's outpatient clinics. Majority of the studies recruited patients with persistent shoulder pain at least for the last 3 months. Moreover, only patients that demonstrated rotator cuff pathology based on clinical and radiological (Magnetic Resonance Imaging (MRI) or Ultrasonography (USG) investigations participated in these studies.

Nearly all studies were 2 armed (intervention vs. control) RCT design, except for one, which compared 3 interventions against control (four-armed study). For this study, only data

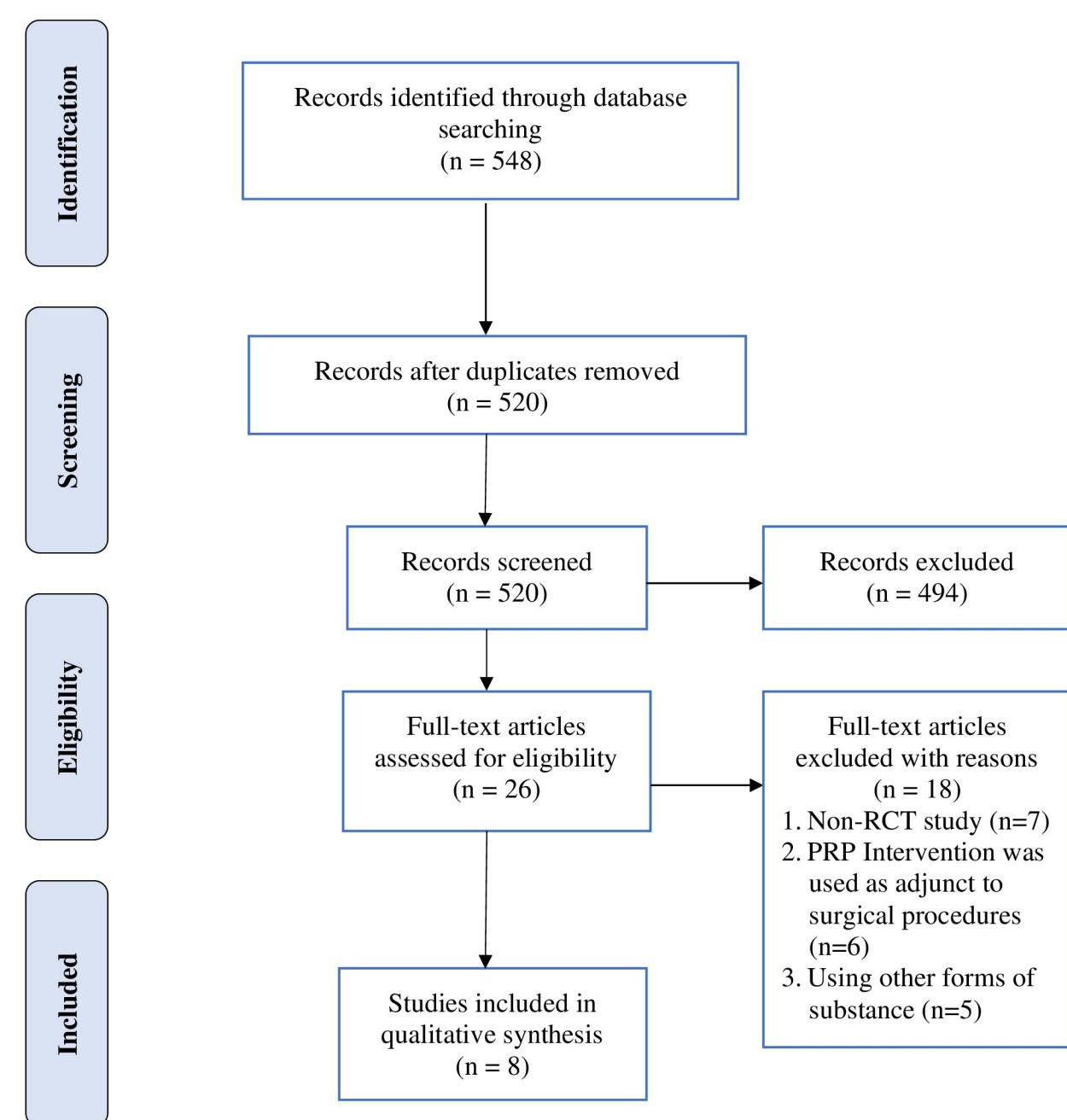

**Fig 1. Flow diagram for selected studies.**

from the PRP and control groups were extracted and analysed [25]. It should also be pointed out that the PRP used in the reviewed studies varies in the technique of preparation (single vs. double centrifugation), use of PRP activating agents during administration, the total volume of PRP injected (ranges from 2 to 4 ml) and the location of infiltration (intralesional vs. subacromial delivery). In addition the frequency of PRP administration (single vs. multiple injections) including duration between repeats also ranges (weekly vs. monthly) and the concurrent use of local anaesthetic agents during injections varies between studies. While most injections were performed under USG guidance [17,25–29], the composition of PRP used were described

**Table 1. Characteristics of selected studies.**

| Author, year, country | Participants, Sample size | Intervention/ Control | Intervention length, Follow-up (FU) from baseline, Retention rate | Intervention characteristics | Outcomes | Significance difference between groups |
|---|---|---|---|---|---|---|
| Cai et. al., 2019, China [25] | • Age: 18–55 years<br>• Partial thickness RCT<br>• Within 6 months diagnosis<br>Sample size: 200 (NS: 50; SH: 50, PRP: 50, SHPRP: 50) | Intervention (IG)<br>• Sodium hyaluronate (SH) group (4 ml)<br>• PRP group (4 ml),<br>• Sodium hyaluronate (SH) (2 ml) + PRP (2 ml) group<br>Control (CG)<br>• Normal saline (NS) group (4 ml) | 12 months<br>FU:<br>1, 3, 6 and 12 months<br>Retention:<br>At 12 months:<br>NS: 94%<br>SH: 88%<br>PRP: 90%<br>SHPRP: 96% | IG: US guided Subacromial injection (4 ml) PRP once a week for 4 wk<br>Total: 4 injections<br>Local anaesthetic administration (1% lidocaine to numb the skin)<br>CG: US guided Subacromial injection (4 ml) NS once a week for 4 wk | Primary:<br>• Constant shoulder score (CSS)<br>Secondary<br>• American Shoulder and Elbow Surgeons Score (ASES)<br>• Visual analogue scale (VAS) | • PRP and SH + PRP groups— significantly higher Constant score and ASES score after the treatments.<br>• Significant differences between the SH + PRP group and the SH or PRP group at 12 months in the Constant, visual analog scale, and ASES scores |
| Centeno et. el., 2020, USA [29] | • Age: 18–69 years<br>• Symptomatic for at least 3 months<br>Sample size: 50 (IG: 25; CG: 25) | Intervention (IG)<br>• US guided BMC (bone marrow concentrate) + PRP injection 2 ml + Standard rehabilitation program<br>Control (CG)<br>• Home rehabilitation program—3 months then cross over allowed | 24 months<br>FU:<br>1, 3, 6, 12, 24 months<br>Retention:<br>At 24 months:<br>IG: 44%<br>CG: 56% | IG: US guided BMC + PRP injection 2 ml + Standard rehabilitation program<br>Single injection<br>CG: Home rehabilitation program —3 months then cross over allowed | Primary:<br>• Disability arm shoulder and hand (DASH)<br>• Numerical pain score (NPS)<br>• Singe assessment numeric evaluation (SANE) | Significant differences outcomes for the BMC treatment compared to exercise therapy at 3 and 6 months for pain, and for function and reported improvement (SANE) at 3 months (p < 0.05). Patients reported a mean 89% improvement at 24 months, with sustained functional gains and pain reduction |
| Kesikburun et. Al., 2013, Turkey [17] | • Age: 18–70 years<br>• > 3 months symptoms<br>Sample size: 40 (IG-20; CG-20) | Intervention (IG)<br>• PRPG–PRP injection (Biomet) 5 ml + Standard rehabilitation program (LR-PRP)<br>Control (CG)<br>• Saline injection 5 ml + Standard rehabilitation program | 6 months<br>FU: 3, 6 weeks, 3, 6 months<br>Retention:<br>At 6 months | IG: US guided injection into the lesion (once)<br>CG: Exercise program initially involved passive range of motion and Codman exercises | Primary:<br>• Western Ontario Rotator Cuff Index (WORC)<br>Secondary:<br>• Shoulder Pain and Disability Index (SPADI)<br>• Pain with Neer impingement sign (VAS)<br>• Passive range of motion using goniometry (flexion, abduction, internal rotation, external rotation) | WORC score: both groups showed significant improvement at all assessment point compared with baseline<br>VAS & SPADI: both groups showed significant improvement at all assessment point compared with baseline<br>PRP no more effective than NS injection |
| Ilhani et. al., 2015, Turkey [23] | • Age: 18–70 y.o<br>Sample size: 70 (IG-35; CG-35) | Intervention (IG)<br>• PRP group injected x 3 (intraarticular)<br>Control (CGT)<br>• Daily home rehabilitation rotator cuff exercise program | 12 months<br>FU: 3/52 & 12 months<br>Retention:<br>At 12 months:<br>IG: 86%<br>CG: 91% | IG: PRP (6 ml) injection into the knee joint– landmark guided<br>CaCl to activate PRP prior to injection<br>Total of 3 injections<br>CG: Physical therapy (continuous mode US) + TENS + ROM + Stretching & strengthening + | Primary<br>• VAS—pain scale<br>Secondary<br>• DASH<br>• Passive range of motion using goniometry (flexion, abduction, internal rotation, external rotation)<br>• Beck depression score | Primary:<br>• Both groups showed significant improvement in ROM and DASH score compared to baseline<br>Secondary:<br>• CG showed significantly higher improvement in ROM and VAS activity compared with IG<br>• IG showed significantly better DASH score than CG |

*(Continued)*

**Table 1.** (Continued)

| Author, year, country | Participants, Sample size | Intervention/ Control | Intervention length, Follow-up (FU) from baseline, Retention rate | Intervention characteristics | Outcomes | Significance difference between groups |
|---|---|---|---|---|---|---|
| Nejati et al., 2017, Turkey [26] | • > 3 months symptoms Sample size: 62 (IG-31; CG-31) | Intervention (IG) PRP group injected x 2 (beginning of study and 1 month after 1st visit) - Control (IG) Exercise therapy group | 6 months FU: 1, 3 and 6 months Retention: At 6 months IG: 65% CG: 58% | IG: Landmark guided injection into tendon and the SASD bursa (2 X) 3 ml injected into tendon (if US demonstrated changes) and 1 ml into SASD bursa (no US guidance) CG: Exercise—supervised once a week (3 months) and performed exercises at home–daily | Primary: • Pain score–VAS Secondary: • Shoulder ROM • Manual muscle test (MMT)—abduction GH joint, adduction scapular • Disability of the arm, shoulder and hand (DASH) • West Ontario rotator cuff index (WORC) | Primary: • Both groups = improvements in VAS Secondary: • Both groups showed significant improvement in ROM in all range • MMT no significant difference • Both groups = improvements in DASH and WORC |
| Rha et al., 2011, South Korea | • Age: 36–79 years • > 6 months symptoms Sample size: 39 (IG-20; CG-19) | Intervention (IG) PRP (3 ml) + home exercise program Control (IG) Dry needling + home exercise program | 6 months FU: 2 weeks, 6 weeks, 3 months, 12 months Retention: At 12 months IG: 80.0% CG: 73.7% | IG: PRP (3 ml), single injection US guided (0.5% lidocaine) + home exercise program CG: Dry needling (2 x—4 weeks apart) + home exercise programme | Primary: • Shoulder Pain and Disability Index (SPADI) • Secondary: Passive Shoulder range of movement (ROM) | Primary: • Both groups showed improvements in SPADI score • More superior clinical effect of PRP than DN at 3 weeks, 3 and 6 months Secondary: • PRP more improvement in shoulder internal rotation and flexion at 3 and 6 months than DN |
| Schwitzguebel et al., 2019, Switzerland [27] | • > 6 months symptoms Sample size: 84 (IG-42; CG-42) | Intervention (IG) PRP group - 2ml PRP injected x2 (1 month apart)—LP-PRP Control (CG) NS–injection x2 (1 month apart) | 12 months FU: 7 and 12 months Retention: At 12 months IG: 97.6% CG: 95.1% | IG: US guided injection, 2 ml of PRP CG: US guided injection of NS | Primary: • Change in lesion volume (MRI) Secondary: • Pain score—visual analog scale (VAS) • Singe assessment numeric evaluation (SANE) • Constant score • American Shoulder and Elbow Surgeons (ASES) score | Primary: • No significant differences between the PRP and control groups in lesion size Secondary: • No significant differences between the PRP and control groups in reduction of pain on—visual analog scale (VAS), improvement in SANE, Constant, ASES |
| Wesner et al., 2016, Canada [24] | • Age 35–60 y.o • Persistent shoulder pain for the last > 3 months Sample size: 7 PRP: 2 NS | Intervention (IG) PRP (Harvest commercial kit) 4 ml injection + daily supervised home-based exercise program Control (CG) NS injection 4 ml into + daily supervised home-based exercise program | 6 months FU: 3 and 6 months Retention: At 12 months IG: 85.7% CG: 50.0% | IG: Landmark guided PRP (4 ml) into degenerative area RC fenestrated + daily supervised home-based exercise program CG: Landmark guided NS 4 (ml) injection into degenerative area RC fenestrated + daily supervised home-based exercise program | Primary • Pain score–VAS Secondary: • DASH score • WORC score • 3. MRI changes | Analysis not done—too small sample size 5 PRP & 1 NS (normal saline)—clinically important DASH improvement 6 PRP & 1 NS—clinically important WORC improvement MRI 5/7 PRP—improved MRI findings NS—no improvement |

RCT = randomised controlled trial, BMI = body mass index, SDS = Standard deviation scores, IG = intervention group, CG = control group, PA = physical activity, SD = standard deviation, BP = blood pressure, DEXA = Dual-Energy X-Ray Absorptiometry, BIA = Bio-impedance analysis.

only in 4 studies [17,23,25,26]. Moreover, in one study, PRP was combined with bone marrow concentrates (BMC) [29]. In contrast, the type of control group was consistent, which included normal saline injections [17,24] or shoulder rehabilitation program [26] or the combination of both [29]. However, in one study, participants in the control group received dry needling injections combined with a home exercise program [28]. The intervention durations ranged from 2 months to 24 months, with the majority [7 of 8] study reported results at 6 months [17,23–28]. Only one study followed up their participants for 2 years [29].

All studies assessed either self-reported pain score using either a visual analogue scale (VAS) or numerical rating pain scale (NRPS) as the primary outcomes; or shoulder functional scores such as Constant Shoulder Score (CSS), American Shoulder and Elbow Score (ASES), Disability of the Arm, Shoulder and Hand (DASH) as the secondary outcome measures. Additionally, 4 studies also measured the shoulder range of movement (ROM) and shoulder muscle tests as part of the secondary outcome measures [17,23,26,28]. All studies assessed outcome measures at several points throughout the study period.

## Study quality

The risk of bias for randomization was low in 6 RCTs [17,25–29]. The allocation concealment was described only in three [37.5%] studies [17,28,29]. Blinding of participants and personnel were described in half of the studies [17,25,27,28]. In all studies, outcome assessors were blinded to participants' allocation group. The risk of incomplete outcome data was low for all studies except for one [29]. The study by Centeno et al. [2020] [29] is an ongoing RCT and they reported findings of the first 6 months of the study. The quality assessment of selected studies is presented in Fig 2. The individual study risk of bias assessments are presented in S1 Fig.

## Outcome of interventions

**Shoulder pain and function.** The primary goals of this review were shoulder pain and post-intervention function. Significant improvement in pain symptoms (VAS score) was observed in both the intervention and control groups relative to the baseline in 6 studies [17,23,25–28]. In 4 studies, participants in the intervention groups showed significantly lower pain scores and higher shoulder function scores compared to controls at the end of the studies [24,25,28,29].

Wesner (2016) [24] reported that participants in PRP showed significant improvement in pain symptoms at 6 weeks compared to the control, but the difference was not statistically significant at 12 weeks and 24 weeks. Centeno et al., 2020 [29] reported significantly lower pain scores as early as 1 month following BMC-PRP shoulder injection and the scores progressively reduced over time throughout the 24 months study duration. Rha et al., 2012 reported a similar trend in a study that compares the effects of PRP injections versus dry needling. The progressive reduction in the SPADI scores was observed in both groups throughout the study; however, participants in the PRP group had significantly lower SPADI scores at 1, 3, and 6-month follow-up [28]. No significant differences in the outcome variables of pain and shoulder function were reported in the remaining 4 studies [17,23,26,27].

We performed meta-analyses to investigate the effect of interventions on shoulder pain. Data from 8 RCTs (n = 976 participants) were pooled to determine effects of PRP on shoulder pain. Since studies varied in the follow-up outcome assessment periods, data from studies with similar follow-up time points were pooled and analysed. Pooled data revealed no significant difference in VAS pain scores between patients treated with PRP and controls at 1 and 3-month follow-up. However, a significant difference in favour of PRP was observed at 12 months post-intervention (SMD = −0.5, CI = −0.7, −0.2, P< 0.001) (Fig 3). Meta-analysis also revealed low heterogeneity at 12 months follow- up of the included studies. Fig 4 depicts an

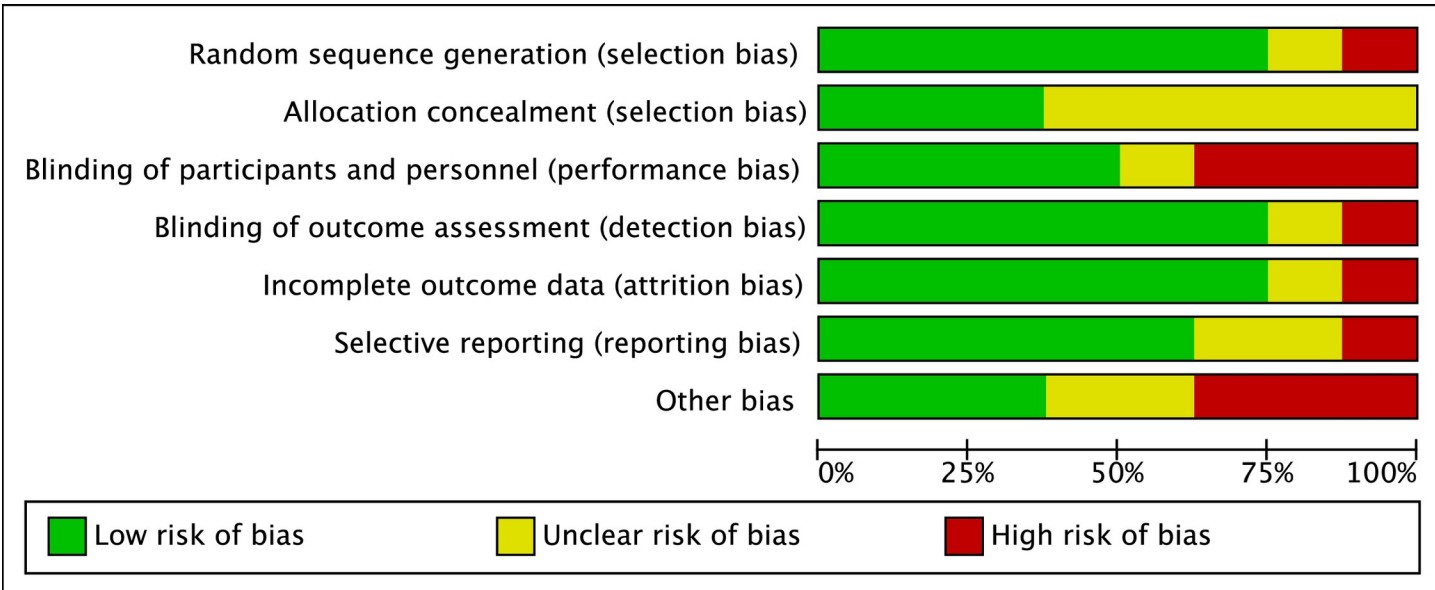

**Fig 2. Quality assessment of selected studies.**

asymmetry funnel plot for the meta-analysis on the effects of intervention on changes in pain score (VAS), suggesting the presence of publication bias.

Based on 3 RCTs pooled meta-analyses of the shoulder range of movement, no significant difference (P>0.05) in changes of shoulder flexion, abduction, internal and external rotation between PRP and controls at 3 and 6 months follow-up was noted.

The meta-analyses of the shoulder functional outcome revealed mixed results. Pooled data were only available for the SPADI and DASH shoulder functional scores. The SPADI shoulder functional score pooled data from 2 studies (n = 228) showed a significant difference in favour of PRP than controls at 3 and 6 months follow-up (Fig 5). In contrast, no significant difference in the DASH scores was found between PRP and controls at all time points (Fig 6). The presence of high heterogeneity between included studies was noted ($I^2$ 90%– 98%).

No funnel plot was generated on the effect of intervention on shoulder range of motion and shoulder functional outcomes since there were only 3 studies included. Moreover, meta-analysis with less than 10 studies lead to low power of analysis to distinguish the chance from real asymmetry [30]. Therefore, the presence or absence of publication bias on these outcomes could not be determined.

### Intervention adverse effects

Adverse effects of intervention were reported in 5 of the included RCTs [17,23,27–29]. Post-injection pain, frozen shoulder and extension of lesion size were reported among participants in both the intervention and control groups by Schwitzguebel et al., 2020. However, more adverse effects were reported among participants treated with PRP compared with normal saline [27]. Temporary post procedural pain (lasting for few days) was the commonest following PRP and BMC-PRP injections [17,23,27–29].

### Discussion

The primary objective of this systematic review was to investigate the clinical effects of PRP and control of rotator cuff injury. Rotator cuff tendinopathy is one of the most prevalent

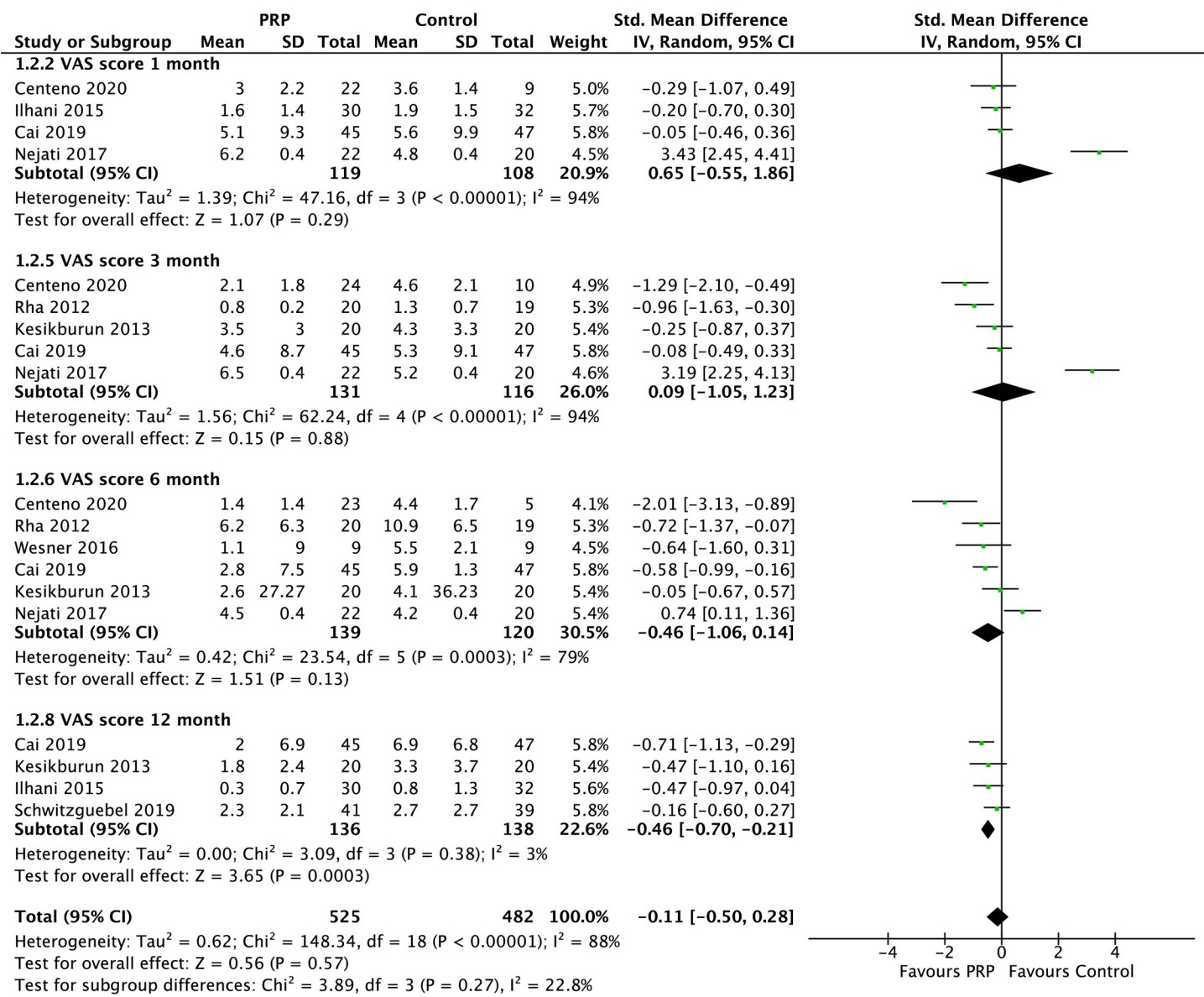

**Fig 3. Forest plot on VAS shoulder pain scores at 1, 3, 6, and 12 months follow-up.**

debilitating musculoskeletal diseases with high incidence and yet the best treatment is still uncertain [2,31].

Our meta-analysis found that there was no difference in the short term (3 weeks) pain symptoms control between PRP and control interventions. PRP injection (s) was significantly better for medium (6 months) and long-term (12 months) pain symptom control. The current findings are consistent with a recent systematic review and meta-analysis, which concluded that PRP was significantly more effective in reducing pain of up to 24 weeks [32].

The clinical advantage of PRP has been reported to occur when the platelet concentrate ranges between 2.5 and 8.0 times greater than whole blood range [33,34]. In this review, only 4 studies described platelet concentration. Three studies [17,23,26] used a platelet concentration of 2 to 4 times higher, while Cai et al., (2018) used PRP with platelet concentration of 8.0 times

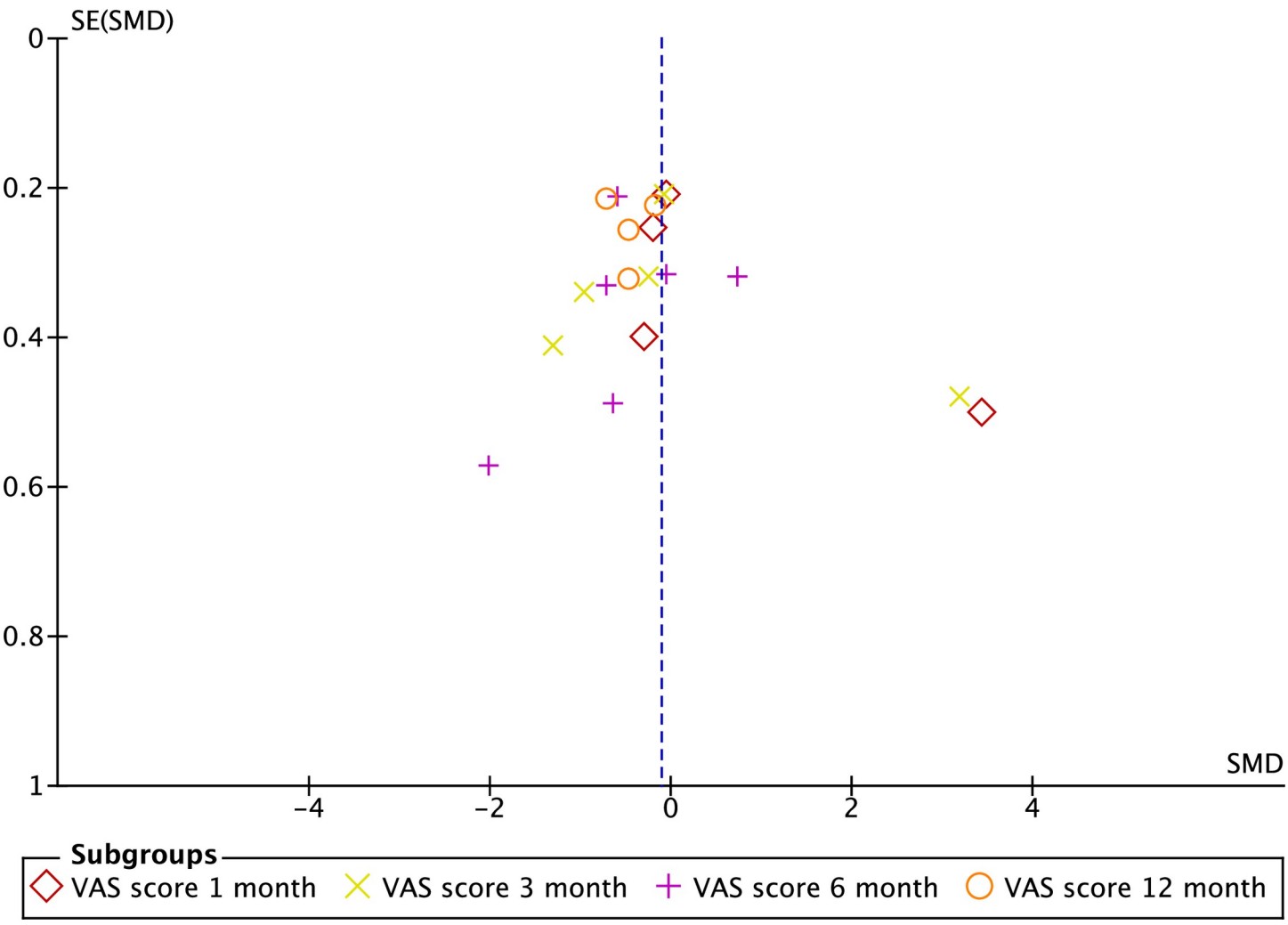

**Fig 4. Funnel plot on VAS pain scores at 1, 3, 6, and 12 months follow-up.**

higher. Despite the higher platelet concentration, Cai reported the most significant reduction in pain was seen in the group receiving combined PRP and sodium hyaluronate [25].

Interestingly, while PRP demonstrated a significant advantage over control for medium and long-term pain symptoms, this effect does not translate consistently with shoulder function scores. Our meta-analysis revealed significantly better SPADI scores among patients treated with PRP than controls at all follow-up time points (3 weeks and 3 and 6 months). However, there was no significant difference in the DASH scores between the PRP and controls at all time points.

SPADI is a basic questionnaire composed of 13 questions that measure the extent of shoulder pain and difficulties in performing everyday life tasks [35]. The DASH questionnaire on the other hand consists of 5 questions on shoulder symptoms and 25 questions related to functional tasks [36]. Several shoulder functional domains are assessed by the DASH questionnaire that are not found in SPADI including occupational related activities, recreational activities and emotional responses to shoulder symptoms. The differences found in the current meta-analysis could relate to the differences in the shoulder function components tested by each

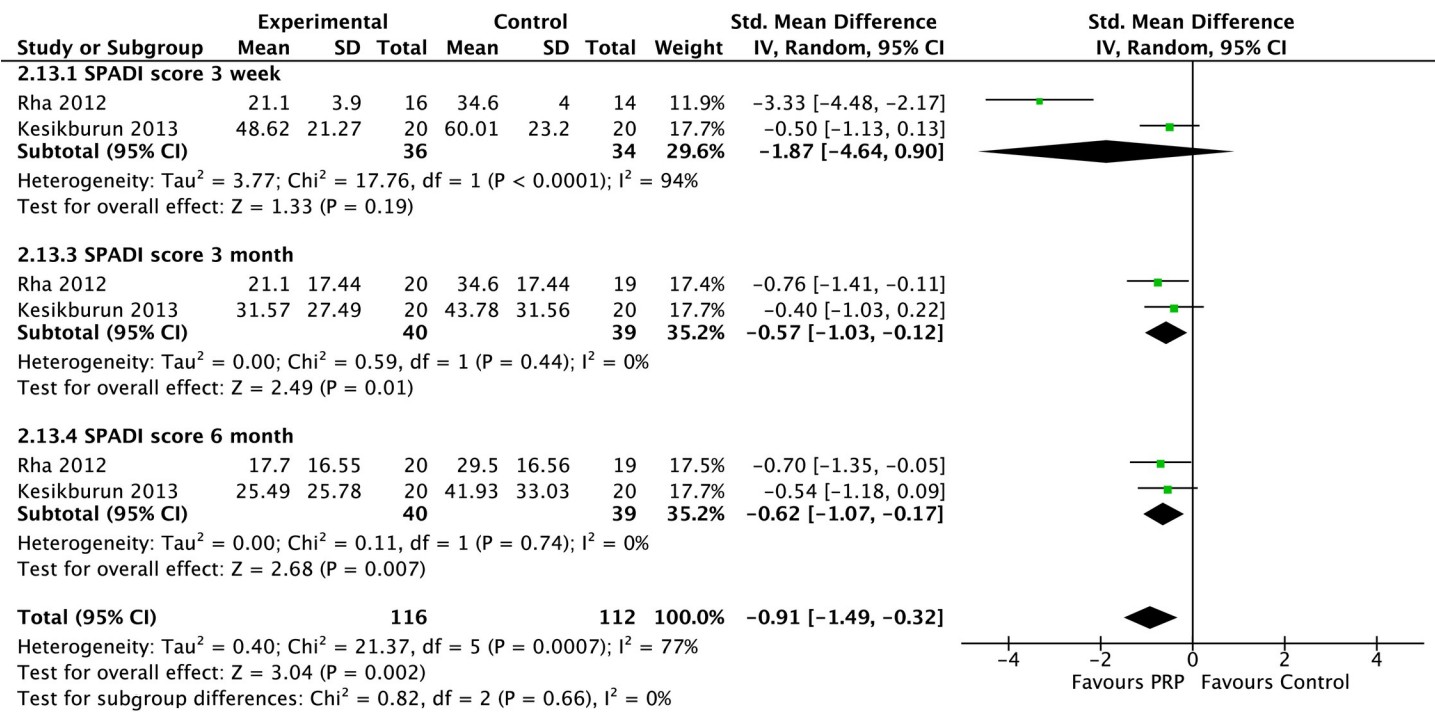

**Fig 5. Forest plot on SPADI shoulder pain scores based on follow-up periods.**

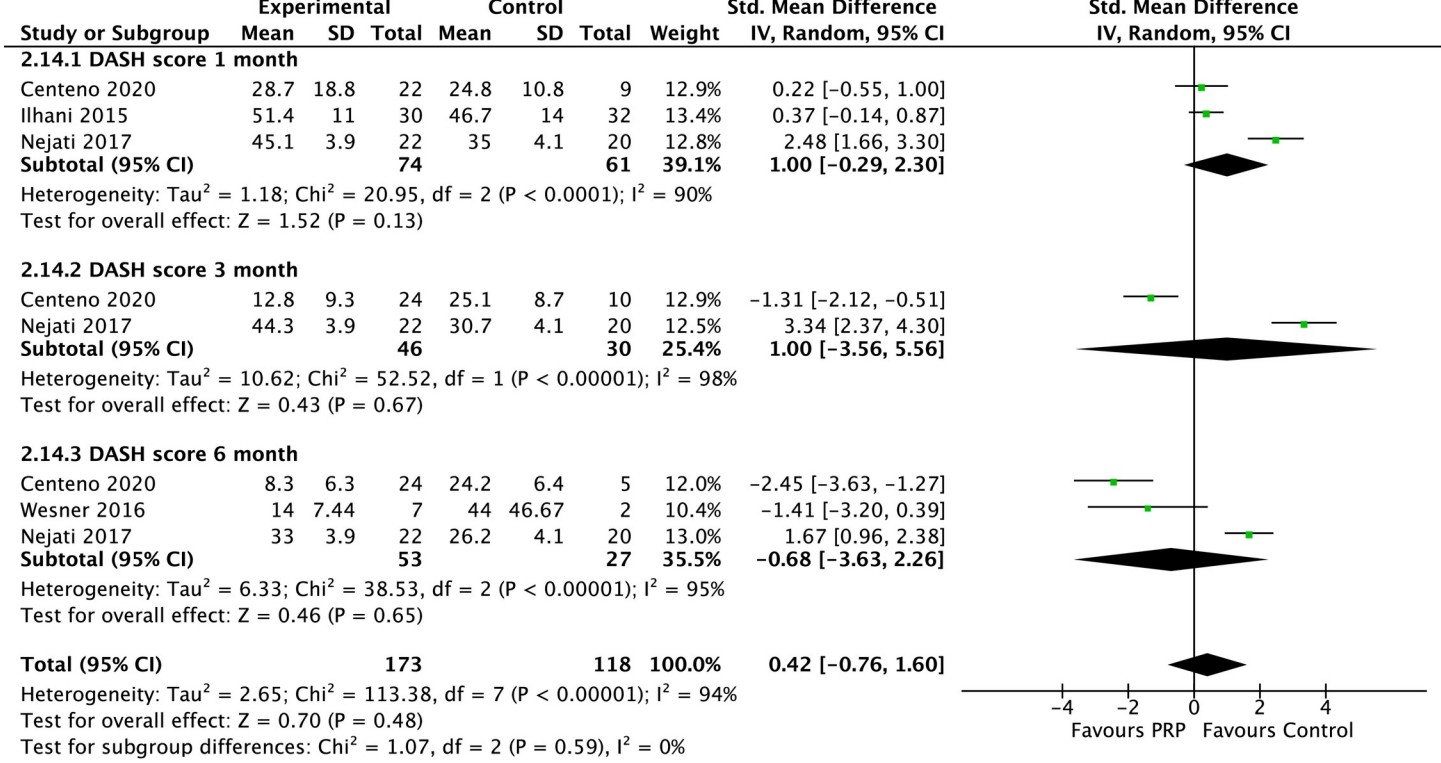

**Fig 6. Forest plot on DASH functional shoulder scores based on follow-up periods.**

questionnaire (unimodal vs. multimodal). Moreover, questionnaires differ in its reliability and validity that ranges from good to excellent [37].

Our findings differed from a recent systematic review and meta-analysis, which documented shoulder function was not significantly improved with PRP injection [32]. In view of the different shoulder functional assessments tools/instruments used in the reviewed RCTs, our meta-analysis results were from pooled and analysed data using the same tool. In contrast, the results from the study by Lin et al., 2020 [32], pooled together all of the different shoulder functional assessment tools.

PRP and prolotherapy were reported to be safe interventions with minimal adverse effects [17,23,27–29]. This review showed a transient increase in pain [less than 3 days] was the most frequent adverse effects reported among PRP treated patients. However, only one study reported other adverse effects such as frozen shoulder and tear extension [27]. Hence, PRP could be considered a safe treatment alternative for rotator cuff tendinopathy.

## Limitation

Several limitations need to be addressed in the review. Although we have used an extensive search strategy, due to limited resources, we restrict only to publications in English. Hence, there is a possibility that some studies were not included. With respect to the allocation concealment and blinding of the participants and personnel, half of the included studies had high or unclear risk of bias. These were, nevertheless, inevitable in view of the nature of the intervention. Variation in PRP intervention including PRP preparation (single or double centrifugations), dose (volume and frequency of administration), injection technique (anatomical or ultrasound guided), and concurrent used of local anaesthetic and the type of PRP was observed in the reviewed RCTs. These could explain the high heterogeneity observed in this meta-analysis. Variation in interventions are not unexpected as PRP used for shoulder tendinopathy is considered a relatively new treatment alternative with inconsistencies in the treatment regime across studies. Therefore, the optimal PRP used for shoulder tendinopathy is yet to be identified.

Finally, the funnel plot showed possible publication bias that may be attributed to studies with small sample size and studies with negative results that were not published. Therefore, the outcomes reported in this meta-analysis should be interpreted with caution.

## Conclusion

This meta-analysis has shown that PRP is safe and more effective for long-term shoulder pain symptoms and shoulder function associated with injury to the rotator cuff. However, more robust studies with a standardised reporting on PRP preparation techniques, PRP class used and injection techniques are recommended. Additionally a standardised, reliable and valid outcome assessment-specific for rotator cuff tendinopathy should be used in future studies.

## Supporting information

**S1 Fig. Individual study risk of bias assessments.**
(TIF)

**S1 File. Prisma checklist.**
(TIF)

**S2 File. Scopus search.**
(DOCX)

## Author Contributions

**Conceptualization:** Mohamad Shariff A. Hamid, Shariff Ghazali Sazlina.

**Data curation:** Mohamad Shariff A. Hamid, Shariff Ghazali Sazlina.

**Formal analysis:** Mohamad Shariff A. Hamid, Shariff Ghazali Sazlina.

**Investigation:** Mohamad Shariff A. Hamid, Shariff Ghazali Sazlina.

**Methodology:** Mohamad Shariff A. Hamid, Shariff Ghazali Sazlina.

**Project administration:** Mohamad Shariff A. Hamid, Shariff Ghazali Sazlina.

**Resources:** Mohamad Shariff A. Hamid, Shariff Ghazali Sazlina.

**Software:** Mohamad Shariff A. Hamid, Shariff Ghazali Sazlina.

**Supervision:** Mohamad Shariff A. Hamid, Shariff Ghazali Sazlina.

**Validation:** Mohamad Shariff A. Hamid, Shariff Ghazali Sazlina.

**Visualization:** Mohamad Shariff A. Hamid, Shariff Ghazali Sazlina.

**Writing – original draft:** Mohamad Shariff A. Hamid.

**Writing – review & editing:** Mohamad Shariff A. Hamid, Shariff Ghazali Sazlina.

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
