## [Decision Letter · Decision Letter 0]

12 Apr 2021

PONE-D-21-06594

Platelet-rich plasma for rotator cuff tendinopathy: A systematic review and

meta-analysis

PLOS ONE

Dear Dr. Sazlina,

Thank you for submitting your manuscript to PLOS ONE. After careful consideration, we feel that it has merit but does not fully meet PLOS ONE’s publication criteria as it currently stands. Therefore, we invite you to submit a revised version of the manuscript that addresses the points raised during the review process.

We look forward to receiving your revised manuscript.

Kind regards,

Sherief Ghozy, M.D., Ph.D. candidate

Academic Editor

PLOS ONE

Journal Requirements:

Thank you for stating the following financial disclosure:

NO - The funders had no role in study design, data collection and analysis, decision to publish, or preparation of the manuscript

2a)         Please clarify the sources of funding (financial or material support) for your study. List the grants or organizations that supported your study, including funding received from your institution.

2b)         State what role the funders took in the study. If the funders had no role in your study, please state: “The funders had no role in study design, data collection and analysis, decision to publish, or preparation of the manuscript.”

2c)          If any authors received a salary from any of your funders, please state which authors and which funders.

2d)         If you did not receive any funding for this study, please state: “The authors received no specific funding for this work.”

Please include your tables as part of your main manuscript and remove the individual files. Please note that supplementary tables (should remain/ be uploaded) as separate "supporting information" files

Please include captions for *ALL* your Supporting Information files at the end of your manuscript, and update any in-text citations to match accordingly. Please see our Supporting Information guidelines for more information: http://journals.plos.org/plosone/s/supporting-information.

Reviewers' comments:

Reviewer's Responses to Questions

**Comments to the Author**

1. Is the manuscript technically sound, and do the data support the conclusions?

Reviewer #1: Yes

Reviewer #2: Yes

2. Has the statistical analysis been performed appropriately and rigorously? 

Reviewer #1: Yes

Reviewer #2: Yes

3. Have the authors made all data underlying the findings in their manuscript fully available?

Reviewer #1: Yes

Reviewer #2: Yes

4. Is the manuscript presented in an intelligible fashion and written in standard English?

Reviewer #1: Yes

Reviewer #2: Yes

5. Review Comments to the Author

Reviewer #1: Rotator cuff (RC) tendinopathy, manifested by shoulder pain and dysfunction, is common, and platelet-rich plasma (PRP) is a popular option in clinics to treat the RC tendinopathy. Although there are many studies to assess the efficacy of PRP treatment on RC pain and function, it is still a matter of debate as to whether PRP treatment is more beneficial than other regular treatment modalities such as physical therapy and corticosteroid injection. The authors of this study performed a systematic review and meta-analysis to assess clinical trials of PRP treatment on RC tendinopathy. They focused on the central question – “How effective are PRP treatment in reducing shoulder pain and improving shoulder function?”

The authors performed extensive search on the literature from 1970 to 2020, found 548 relevant articles, and eventually identified 8 qualified articles, which were randomized controlled trials (RCT). The criteria for the selection of the previous studies are clearly defined.

Then they performed meta-analysis on these eight RCT studies to assess the study quality, the primary outcomes – VAS score, or NRPS score; and secondary outcomes - CSS, ASES, and DASH. The main finding of this meta-analysis is that there was no improvement in pain from PRP treatment compared to control interventions in early treatment time point (3 weeks), but significant improvement in pain was revealed at 6 and 12 months after PRP treatment.

Overall, the authors performed this meta-analysis of previous RCTs rigorously and as a result, their findings are deemed to be reasonable. The manuscript is well written with all necessary details provided so that the readers can evaluate the quality of their analysis of literature in general and those RCTs included in this report in particular.

In the “Limitation” section, the authors state that because of small sample size and lack of - CSS unpublished negative studies, “the outcomes reported in this meta-analysis should be interpreted with caution.” This reviewer agrees with this statement completely. In fact, even the highly selected 8 RCTs were not homogenous at all (I square > 90%, page 19). The heterogeneity among the studies in this report can also include PRP-related factors - the technique of preparation, use of activating agents, volume of PRP injected, the frequency of PRP injection, and those patient-related factors such as different stages of RC tendinopathy. The authors may wish to discuss these factors in more details so that readers can appreciate the complexity of a clinical trial more with PRP treatment on RC tendinopathy.

Reviewer #2: The article is of scientific interest and in line with the aims of the journal. The authors guidelines have been respected and the manuscript does not require a revision of the English language by a native speaker. The literature search was adequately carried out and the results of the studies were described in full.

The only concern are the tables and figures that in the document I have are of very bad quality and it is not possible for me to judge them.

Therefore you need to increase the quality of tables and figures before proceeding with publication.

6. PLOS authors have the option to publish the peer review history of their article (what does this mean?). If published, this will include your full peer review and any attached files.

Reviewer #1: No

Reviewer #2: No

---

## [Author Response · Author response to Decision Letter 0]

18 Apr 2021

Reviewer #1

1. Rotator cuff (RC) tendinopathy, manifested by shoulder pain and dysfunction, is common, and platelet-rich plasma (PRP) is a popular option in clinics to treat the RC tendinopathy. Although there are many studies to assess the efficacy of PRP treatment on RC pain and function, it is still a matter of debate as to whether PRP treatment is more beneficial than other regular treatment modalities such as physical therapy and corticosteroid injection. The authors of this study performed a systematic review and meta-analysis to assess clinical trials of PRP treatment on RC tendinopathy. They focused on the central question – “How effective are PRP treatment in reducing shoulder pain and improving shoulder function?”

The authors performed extensive search on the literature from 1970 to 2020, found 548 relevant articles, and eventually identified 8 qualified articles, which were randomized controlled trials (RCT). The criteria for the selection of the previous studies are clearly defined.

Then they performed meta-analysis on these eight RCT studies to assess the study quality, the primary outcomes – VAS score, or NRPS score; and secondary outcomes - CSS, ASES, and DASH. The main finding of this meta-analysis is that there was no improvement in pain from PRP treatment compared to control interventions in early treatment time point (3 weeks), but significant improvement in pain was revealed at 6 and 12 months after PRP treatment.

Overall, the authors performed this meta-analysis of previous RCTs rigorously and as a result, their findings are deemed to be reasonable. The manuscript is well written with all necessary details provided so that the readers can evaluate the quality of their analysis of literature in general and those RCTs included in this report in particular.

In the “Limitation” section, the authors state that because of small sample size and lack of - CSS unpublished negative studies, “the outcomes reported in this meta-analysis should be interpreted with caution.” This reviewer agrees with this statement completely. In fact, even the highly selected 8 RCTs were not homogenous at all (I square > 90%, page 19). The heterogeneity among the studies in this report can also include PRP-related factors - the technique of preparation, use of activating agents, volume of PRP injected, the frequency of PRP injection, and those patient-related factors such as different stages of RC tendinopathy. The authors may wish to discuss these factors in more details so that readers can appreciate the complexity of a clinical trial more with PRP treatment on RC tendinopathy.

Response: Thank you for your encouraging comments. We have elaborated on the potential factors that could have contributed to the heterogeneity of the RCTs reviewed as suggested. Kindly refer to Page 23, Lines 199 - 204.In addition, we have included a discussion as suggested on these factors (Page 29, Lines 340 – 347)

Reviewer # 2

1. The article is of scientific interest and in line with the aims of the journal. The authors guidelines have been respected and the manuscript does not require a revision of the English language by a native speaker. The literature search was adequately carried out and the results of the studies were described in full.

The only concern are the tables and figures that in the document I have are of very bad quality and it is not possible for me to judge them.

Therefore you need to increase the quality of tables and figures before proceeding with publication. 

Response: Thank you for the encouraging comments. We have improved on the quality of the tables and figures for better clarity.

---

## [Editor Report · Decision Letter 1]

21 Apr 2021

Platelet-rich plasma for rotator cuff tendinopathy: a systematic review and

meta-analysis

PONE-D-21-06594R1

Dear Dr. Sazlina,

We’re pleased to inform you that your manuscript has been judged scientifically suitable for publication and will be formally accepted for publication once it meets all outstanding technical requirements.

Kind regards,

Sherief Ghozy, M.D., Ph.D. candidate

Academic Editor

PLOS ONE
---

## [Editor Report · Acceptance letter]

28 Apr 2021

PONE-D-21-06594R1 

Platelet-rich plasma for rotator cuff tendinopathy: a systematic review and meta-analysis 

Dear Dr. Sazlina:

I'm pleased to inform you that your manuscript has been deemed suitable for publication in PLOS ONE. Congratulations! Your manuscript is now with our production department. 

Kind regards, 

on behalf of

Dr. Sherief Ghozy 

Academic Editor

PLOS ONE